



# An effective parameter optimization with radiation balance constraint in CAM5 (version 5.3)

Li Wu[1], Tao Zhang[3], Yi Qin[2], Wei Xue[1, 2]

[1]Department of Computer Science and Technology, Tsinghua University, Beijing, 100084, China

[2]Ministry of Education Key Laboratory for Earth System Modeling, Department of Earth System Science, and, Joint Center for Global Change Studies, Tsinghua University, Beijing, 100084, China
[3]Brookhaven National Laboratory, New York, 11973, USA

*Correspondence to*: Wei Xue (xuewei@tsinghua.edu.cn)

**Abstract.** Uncertain parameters in physical parameterizations of General Circulation Models (GCMs) greatly impact model
performance. In recent years, automatic parameter optimization has been introduced for tuning model performance of GCMs, but most of the optimization methods are unconstrained optimization methods under a given performance indicator. Therefore, the calibrated model may break through essential constraints that models have to keep, such as the radiation balance at top of model. The radiation balance is known for its importance in the conservation of model energy. In this study, an automated and efficient parameter optimization with the radiation balance constraint is presented and applied in the Community Atmospheric
Model (CAM5) in terms of a synthesized performance metric using global means of radiation, precipitation, relative humidity, and temperature. The tuned parameters are from the parameterization schemes of convection and cloud. The radiation constraint is defined as the deviation of the net longwave flux at top of model (FLNT) and the net solar flux at top of model (FSNT) less than 1 W m$^{-2}$. Results show that the synthesized performance under the optimal parameters is 6.3 % better than the control run (CNTL) as well as the radiation imbalance is as low as 0.1 W m$^{-2}$. The proposed method provides the insight
for physics-guided optimization under the premise of a profound understanding of models and it can be easily applied to optimization problems with other prerequisite constraints in GCMs.

## 1 Introduction

The subgrid-scale physical processes in General Circulation Models (GCMs) are represented by parameterization schemes, which may exist with several uncertain parameters. Inappropriate parameters can seriously affect the overall performance of
the model and may result in breaking physical mechanisms that models have to address. The Intergovernmental Panel on Climate Change Fourth Assessment Report (IPCC AR5) pointed out that studies on parameter uncertainty are critical to



improve climate simulation capabilities (Mastrandrea et al., 2011). Bauer et al. (2015) also indicated that small errors in the physical parameterization schemes could lead to large-scale systematic errors. Traditionally, to achieve better performance, the uncertain parameters are tuned based on the experience of model experts and statistical analysis. This is a labor-intensive job and the tuning results are difficult to achieve local or global optimality in complex climate models.

To efficiently reduce parameter-introduced uncertainty, quite a few automated parameter calibration methods have been proposed. These calibration methods can be categorized into two types. One attempts to obtain the probability distributions of the parameters by likelihood and Bayesian estimation methods. Cameron et al. (1999) exploited the generalized likelihood uncertain estimation to obtain parameters ranges with a specific confidence level. An adaptive Markov Chain Monte Carlo (MCMC) was used to calibrate the uncertain parameters in the ECHAM5 climate model (Järvinen et al., 2010). Edwards et al.

(2011) proposed a simplified procedure of Bayesian calibration to make a quantification of uncertainty in climate forecasting. This type of method has also been successfully applied to the CAM3.1 model and the third Hadley Centre Climate Model (HadCM3) (Jackson et al., 2008; Williamson et al., 2013).

The other method is to adjust parameters by using optimization methods to minimize the errors between model simulations and observations, which are formulated with a given performance indicator. Many intelligent evolutionary optimization

algorithms were applied to model tuning. For examples, both simulated stochastic approximation annealing (SSAA) (Yang et al., 2013) and multiple very fast simulated annealing (MVFSA) (Zou et al., 2014) were used for uncertainty quantification and parameter calibration.

Both methods can consider the interaction of parameters, achieve automatic optimization, and avoid the subjectivity and experientiality of manual calibration. However, they also share high computation cost challenges due to the hundreds and

thousands required simulations. This is usually unacceptable, especially for high-resolution climate models. To overcome the computational issues, the surrogate model, which is a way to replace the real climate model with a cheaper statistical model for faster optimization, has been recently introduced. Applications of these methods in climate models include the works presented by Neelin et al. (2010) and Wang et al. (2014). However, training a precise surrogate model for a complicated climate model such as CESM is very challenging. Moreover, capturing the climatic characteristics of extreme events is difficult for

the cheap statistical model. To make it possible to optimize parameters efficiently and quickly in the complex and highly nonlinear earth system models, an improved simplex algorithm was presented by Zhang et al. (2015). This method can overcome the shortcomings of the traditional simplex downhill method, and the computing efficiency of the algorithm is improved compared with evolutionary optimization algorithms.

The application of various automatic parameter optimization methods in climate models has gradually received more attention,

however, the above optimization algorithms mentioned are mostly unconstrained, and they lack emphasis on the physical mechanisms of the model itself. This paper takes radiation balance as an example, according to the Earth's energy conservation theory, the absorbed solar radiation is approximately equal to outgoing longwave radiation at the top of model. Forster et al.



(2007) proposed that radiative balance is critical to the Earth's system, and the bias of radiation has a big impact on the change of surface temperature. Hourdin et al. (2017) pointed out that a 1 W m$^{-2}$ change in global energy balance may result in a global mean surface temperature change of 0.5 to 1.5 K in coupled simulations. Additionally, Wild (2008) indicated that radiation biases in the GCMs may influence climate sensitivity, thus possibly distorting the prediction of future climate conditions. Lin

et al. (2010) showed the extreme importance of climate energy imbalance and stressed that long-term high-precision measurements of TOA radiation are necessary.

Radiation balance is critical for GCMs, but its deviation can still exceed 1 W m$^{-2}$ in some CMIP5 models (Smith et al., 2014). To better understand this problem, many studies have tried to determine the cause of radiation deviation by analyzing the influence of uncertain parameters and making corresponding adjustments. Zhao et al. (2013) concluded that cloud

microphysics and emission related parameters have statistically important impacts on the global mean net radiation flux. Qian et al. (2015) indicated that some parameters in cloud microphysics and convection are very sensitive to net radiation flux. The improvement of the simulation performance of the climatology and variability based on the radiation balance is very meaningful. However, the constrained optimization methods used to calibrate parameters with physical constraints in climate models remain to be further studied. Cheng et al. (2018) showed that penalty functions and separation of objective and

constraint methods are popular for solving constrained problems. Penalty methods encourage search toward feasible regions by increasing the objective function value with a penalty value for the points that violate the constraints. The exterior penalty method is relatively easy to implement, and it can be widely used in various algorithms. The separation of objective and constraint is commonly used by transforming constraints into to objectives, but it limited by the convergence of the multi-objective algorithms when the optimization problem is high computational cost.

The purpose of this paper is to propose an effective constrained optimization method and demonstrate its feasibility in the calibration of uncertain parameters under the premise of ensuring the balance of radiation. And this paper is organized as follows. Section 2 describes the details of the model and experimental design. Section 3 introduces the new constrained parameter calibration method. Evaluations and analysis of the optimization results are presented in Section 4. The last section contains the conclusion and discussion.

**2 Model and Experiment**

**2.1 Model Description**

The model used in this study is CAM5 (release v5.3),which is the atmospheric component of the Community Earth System Model (CESM 1.2.1). The dynamical core uses the finite-volume method developed by Lin and Rood (1996) and Lin (2004). More details on CAM5 can be found in the work of Neale et al. (2010). Deep convection is handled by a parameterization

scheme developed by Zhang and McFarlane (1995) with the further modifications of Richter and Rasch (2008), as well as





Neale et al. (2008). The original parameterization of stratiform cloud microphysics is handled by Morrison and Gettelman (2008). Modifications of ice nucleation and ice supersaturation can be found in Gettelman et al. (2010). The parameterization of fractional stratiform condensation is described by Zhang et al. (2003), as well as Park et al. (2014). Radiation scheme uses the Rapid Radiative Transfer method for GCMs (RRTMG) (Mlawer et al., 1997; Iacono et al., 2008).

## 2.2 Experiment Design

Table 1 shows the parameters to be adjusted, the ranges, and the default values. These parameters were identified as sensitive to cloud and convection process in previous studies. Qian et al. (2018) showed that deep convection precipitation efficiency zmconv_c0_lnd and zmconv_c0_ocn have significant impact on the variance of shortwave cloud forcing (SWCF) over the land and ocean, respectively. Thresholds of relative humidity for high and low stable clouds (cldfrc_rhminh and cldfrc_rhminl) are considered as the important parameters to cloud and radiation (Zhang et al., 2018). In addition, the relative humidity threshold for low clouds is also one of the strongest parameters impacting the global mean precipitation and makes a huge contribution to the TOA net radiative fluxes in CAM5 (Qian et al., 2015). The timescale for consumption rate of deep CAPE (zmconv_tau) is identified as the most sensitive parameter to the convective precipitation in Zhang-McFarlance by Yang et al. (2013). The cloud ice sedimentation velocity (cldsed_ai) has a significant effect on cloud radiative forcing (Mitchell et al., 2008), and it has been identified as the second most influential parameter in climate (Sanderson et al., 2008). The ranges of these parameters are referenced to previous studies (Qian et al., 2015; Zhang et al., 2018).

The output variables used to synthesize a performance indicator are longwave cloud forcing (LWCF), SWCF, precipitation (PRECT), humidity at 850 hPa (Q850) and temperature at 850 hPa (T850), shown in Table 2. The observations of LWCF and SWCF are from CERES-EBAF (Clouds and the Earth's Radiant Energy System-Energy Balanced and Filled, Loeb and Coauthors, 2014). PRECT is from GPCP (Global Precipitation Climatology Project, Adler et al., 2003). And Q850 and T850 are from ERA-Interim, which was produced by the ECMWF (Dee et al., 2011).

In this study, we use 1.9° latitude × 1.9° longitude resolution CAM5 with 30 vertical layers. Each simulation is a 5-year AMIP from 2000 to 2004 with the observed climatological sea surface temperature (SST) and sea ice (Rayner et al., 2003). The simulations in the last 3 years are used to evaluate the synthesized performance metric and constraint.

## 3 Method

A constrained automatic optimization method is proposed based on Zhang et al. (2015). The synthesized metric used to evaluate the performance of overall simulation skills are shown in Eq. (3):

$$(\sigma_m^F)^2 = \sum_{i=1}^{I} w(i)(x_m^F(i) - x_o^F(i))^2 \tag{1}$$

$$(\sigma_r^F)^2 = \sum_{i=1}^{I} w(i)(x_r^F(i) - x_o^F(i))^2 \tag{2}$$



$$\chi^2 = \frac{1}{N^F} \sum_{F=1}^{N^F} \left(\frac{\sigma_m^F}{\sigma_r^F}\right)^2 \tag{3}$$

$(\sigma_m^F)^2$ represents a criterion for the simulation skill of the models with modified parameters, $(\sigma_r^F)^2$ is an evaluation of the default experiment simulation skill. If the indicator $\chi^2$ is less than 1, this means that the simulation with tuned parameters is better than the CNTL. The smaller the index, the better performance of model. The model outputs are represented by $x_m^F(i)$,

and $x_o^F(i)$ denotes the corresponding reanalysis or observation data. The expression $x_r^F(i)$ represents model outputs from the CNTL. The weight of the different grids on the sphere is represented by $\omega$. $I$ denotes the total number of grids in model. The number of the output variables in the performance index is represented by $N^F$.

The radiation balance is defined as the absolute value of the difference between net solar flux (FSNT) and net longwave flux (FLNT) in climatology at the top of the model less than 1 W m$^{-2}$, which is the maximum deviation of radiation observations

in the decade before 2014 (Trenberth et al., 2014).

Coupled with the radiation balance constraint, the optimization problem of this study can be expressed as Eq. (4) (5):

$$\min \chi^2 \tag{4}$$

$$subject.to.\ ABS(FSNT_m - FLNT_m) < 1 \tag{5}$$

Converting the unconstrained problem into a constrained problem using the penalty function method, it can be transformed

into the augmentation function as Eq. (6):

$$F(x) = \chi^2 + \beta * ABS(FSNT_m - FLNT_m) \tag{6}$$

The penalty factor $\beta$ for the constraint in Eq. (6) is chosen to be 10000 if the constraint in Eq. (5) is not satisfied, otherwise it is equal to 0. The purpose of this choice is to optimize the search space to avoid the possibility of radiation imbalance. This penalty function method is easy and effective when dealing with this tightly constrained optimization.

We use the improved simplex downhill method, proposed by Zhang et al. (2015), to optimize the augment function. Firstly, the single parameter perturbation sample method (SPP) is used to obtain several better initial values while ensuring that the initial geometry of simplex downhill is well-conditioned. The initial value preprocessing mechanism ensures that algorithm starts from a good basis. This is important for the simplex algorithm, which is easy to fall into local optimum. Next, the simplex downhill algorithm is applied to search for better performance.

$F(x)$ gradually converges as shown in Fig. 1. There are some cases in which the radiation balance is not satisfied at the beginning of optimization. However, as the iteration step increases, the search space of the algorithm is constrained within the feasible range. The goal is then to make the synthesized performance metric smaller. In addition, a comparative experiment with unconstrained algorithms is done to verify our doubts about unconstrained methods. Fig. 2 shows the performance indices and radiation deviations corresponding to the first 15 solutions after the two algorithms converge respectively. The constrained

optimization algorithm can find solutions that are more radiation balanced, however, the final solution metrics are not as good





as the unconstrained optimization algorithm. Compared to the CTNL experiment, we can find quite a few solutions with better metrics and smaller radiation biases.

## 4 Result

### 4.1 The optimal model

The best uncertainty parameters obtained by unconstrained optimization method optimize the overall performance of the simulation by 10.1 %, but they have a radiation deviation up to 3.8 W m$^{-2}$. When considering the converged constrained optimization algorithm, the optimal parameters can improve the model performance by 6.3 %, and the radiation imbalance is as low as 0.1 W m$^{-2}$. The corresponding results of the optimal solutions with the two methods are shown in Table 3. Both unconstrained optimization and constrained optimization can further improve the simulation performance, but unconstrained

optimization may encounter an optimal solution that does not satisfy the radiation balance, thus leading meaningless optimization. The optimization results discussed below are based on the proposed constrained optimization method.

The optimization of each output variable is shown in Table 4. In addition, a Taylor diagram is used to estimate the model performance through the standard deviation and correlation (Fig. 3). By combining the results of Table 4 and Figure 3, it can be concluded that SWCF and Q850 receive most optimization, as they achieve a better performance index. Also, compared to

the default experiment, their standard deviations have improved. Table 5 shows the standard deviations of the variables, which are important for the model but not used as evaluation criteria. It is noteworthy that they are also close to the default experiment. For a more comprehensive analysis of the spatial variation of the output variables, the zonal distribution of the difference between EXP/CNTL and observations of all metric variables are shown in Fig. 4. SWCF and Q850 have been obviously improved over low and middle latitudes, but the changes of PRECT and T850 are not particularly notable. Further, LWCF

only showed significant improvement near the equator, and it slightly deteriorated over the middle and high latitudes.

### 4.2 Interpretation of the results

The optimized parameters values are provided in the "Constrained tune" column of Table 1. The deep convection precipitation efficiency over land and ocean are reduced relative to the default values. The timescale for consumption rate of CAPE for deep convection is smaller than the default value, and both relative humidity thresholds for high and low clouds are increased.

Additionally, the sedimentation velocity of cloud ice is larger. Next, we will explain how the changes in these parameters are related to the results of the simulations.

The relative humidity threshold for low clouds is larger in optimization experiments than the default value, which will obviously lead to the decrease of low cloud fraction. The decreased low cloud fraction is consistent with the increase of SWCF.





The CNTL experiment has excelled in simulating the spatial distribution of SWCF (Fig. 5c), but it has a negative bias over the ocean in the low latitudes, where the improvement is significant in the optimal experiment.

The zonal mean specific humidity at 850 hPa is significantly improved, and its spatial distribution is presented in Fig. 6. In the optimal experiment, the atmosphere is drier in tropics and middle latitudes, which is closer to the observation than the CNTL experiment. Meanwhile, the middle to low troposphere is also slightly drier in these areas (Fig. 7), which may be related to the increased convective precipitation. A quasi-equilibrium closure is used in the deep convection scheme in CAM5, which is based on CAPE. The adjustment timescale represents the denominator of the cloud bottom convective mass flux. When the time scale is shorter with unchanged cloud bottom convective mass flux, the convective mass flux is larger, and the convective precipitation increases. Additionally, compared to the CNTL experiment, the lower troposphere gets warmer and the middle troposphere is colder, which exacerbates the instability of the temperature structure (Fig. 8) and leads to more convective precipitation. The spatial distribution of convective precipitation over the tropics where convection occurs most frequently can be seen in Fig. 9. The increase in convective precipitation may be related to the decrease in specific humidity at 850 hPa. However, the increase of total precipitation is not particularly significant, which is dominated by the changes in convective precipitation. The main reason is likely associated with the decreased precipitation efficiency parameters, which could reduce the convective precipitation as a compensation. Therefore, the decreases of precipitation efficiency partially offset the precipitation change caused by tau and temperature structure.

It is difficult for all variables to be optimized, due to the strong interaction among parameters and the complex relationship among output variables. The simulations of T850 between optimal and CNTL experiments are very similar. It is likely the results of the combined effects of all relevant parameterizations. In the optimal experiment, LWCF is closer to the observation in the tropics, but it becomes slightly smaller at middle to high latitudes compared to the CNTL experiment, which implies the larger bias. The relative humidity threshold for high clouds and the sedimentation velocity of ice crystals are correspondingly increased, and both of them would lead to the reduction in high clouds. High cloud fraction changes compared to the CNTL experiment can be seen in Fig. 10c. The reduced high cloud is consistent with the reduction in LWCF. Cloud changes also inevitably affect SWCF. It can be seen the middle cloud has increased relative to the default experiment (Fig. 10c), and the increase of the middle cloud may be related to the decrease of precipitation efficiency over ocean.

In conclusion, the increase in SWCF is consistent with the decrease of cloud fraction for the sake of larger relative humidity threshold of low clouds. Changes in the Q850 are related to increased convective precipitation. Precipitation only slightly increases in the tropics and the global total precipitation has changed very little, which is related to comprehensive effect of the changes of the convection adjustment timescale, the precipitation efficiency parameter, and the vertical temperature structure. T850 simulated by the optimization experiment is similar to the default experiment. The reduced LWCF is related to the decreased high clouds caused by the increased relative humidity threshold for high clouds and the increased sedimentation velocity of ice crystals.





## 5 Conclusion and Discussion

Radiation balance is a crucial factor for the long-term energy balance of GCMs, but it has not received enough attention in automatic parameter optimization. First of all, this paper points out that the previous parameter optimization algorithms do not consider radiation balance as a necessary condition, and the obtained optimization parameters are likely to break this important

physical constraint, which may lead to unacceptable calibrated parameters. Thus we propose an efficient constrained automatic optimization algorithm to calibrate the uncertainty parameters in CAM5 with the constraint of the absolute value of the difference of net solar flux and net longwave flux at the top of the model (less than 1 W m$^{-2}$). In the parameter calibration, we use the comprehensive performance with five fields of LWCF, SWCF, PRECT, Q850 and T850 as the performance indicator. We choose the uncertain parameters in cloud and convection parameterizations, including the deep convection precipitation

efficiency over land and ocean, thresholds of relative humidity for stable high and low clouds, the timescale for consumption rate deep CAPE, and the ice falling speed. Each simulation in our optimization experiments is a 5-year AMIP experiment forced with prescribed seasonal climatology of SST and sea ice.

The optimal parameters found by our method can increase the overall performance of the model by 6.3 %, and the radiation imbalance is as low as 0.1 W m$^{-2}$. The most optimized variables are SWCF and Q850. The increase in SWCF is consistent

with the decrease of cloud water due to larger relative humidity threshold value for low clouds. The reduction of the Q850 in the troposphere may be related to the increase of convective precipitation. The change in global total precipitation is not particularly obvious, which is likely the comprehensive effect of the changes of convection adjustment timescale, the precipitation efficiency parameter, and the structure of temperature over troposphere. The change of T850 is very small, and the result is slightly better than that of the default experiment. Meanwhile, under the constraint of energy balance, LWCF has

deteriorated in the middle and high latitudes. This also reflects some issues that may exist in the structure of model.

The unconstrained optimization methods calibrate the uncertain parameters in climate models without consideration of principles that model have to hold, this creates challenges in maintaining the physics constraints and improving the structure of models. Perhaps a more physics-guided optimization is a better way to understand the principles of climate systems and best use these principles in tuning processes. In the future, we will apply this method to coupled models, where the radiation

imbalance has a more significant impact on long-term simulation stability. In addition, we will also try to introduce more constraints, such as the surface energy balance, into automatic parameter calibration.

*Code and data availability.* The code of our algorithm, the observations and the related scripts can be found at https://github.com/wuli-qhu/Constrained-tuning-in-CAM5. The source code of CAM5.3 are available from

http://www.cesm.ucar.edu/models/cesm1.2/. If you have any problem, please feel free to contact us (wulitianyi@gmail.com).





*Author contributions.* LW, TZ proposed the tuning method, LW, YQ, WX designed the metrics and the constraint. YQ and LW evaluate the optimal results. LW, TZ, WX and YQ wrote the paper.

*Competing interests.* The authors declare that they have no conflict of interest.

*Acknowledgements.* This work is supported by the National Key R&D Program of China (grant nos. 2017YFA0604500 and 2016YFA0602100).

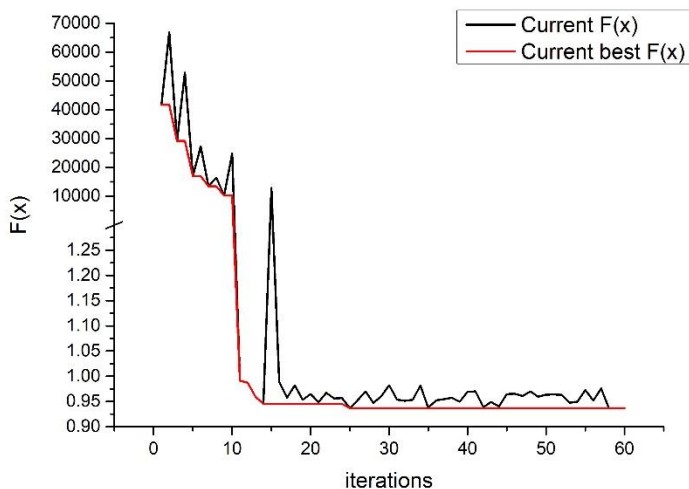

10 **Figure 1. The change of augmentation function F(x) across the optimization iterations. The x axis is the number of iterations. The y-axis is the value of F(x) in Eq. (6). The black line shows the value of F(x) in a given iteration step, while the red line shows the best F(x) value up to the current iteration step.**

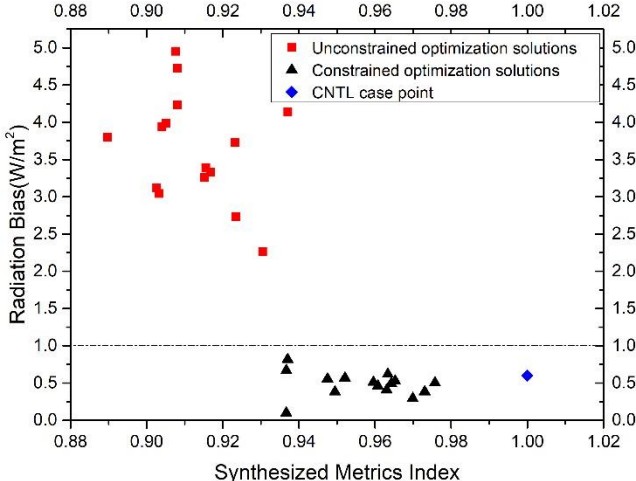

**Figure 2. Comparison of results between the constrained optimization algorithm and the unconstrained optimization algorithm. The 15 red squares and 15 black triangles are optimized solutions found by the unconstrained optimization algorithm and constrained algorithms respectively. The blue diamond is the result of the CNTL experiment. The x axis is the synthesized metric index in Eq. (3). The y axis is the radiation bias at top of model.**



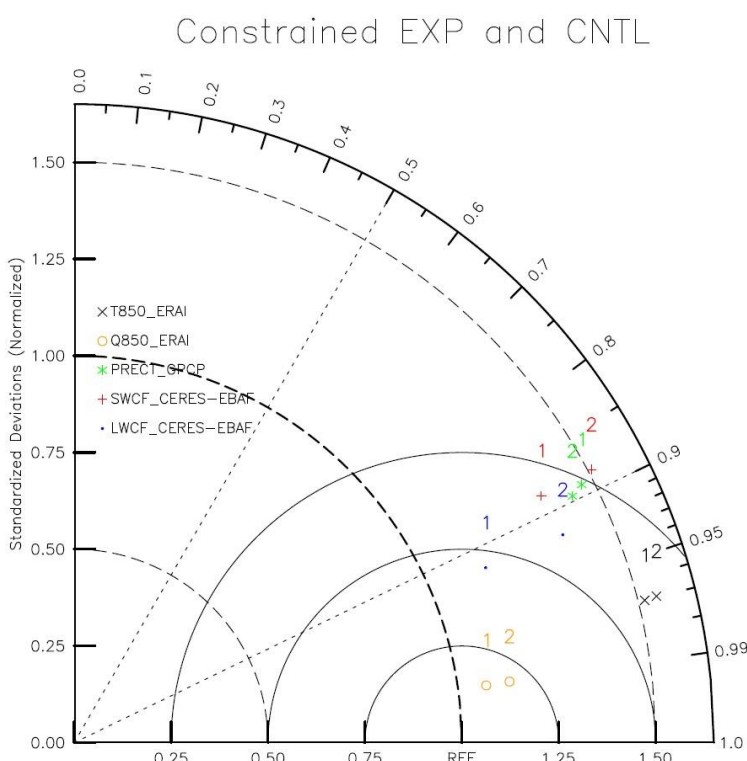

**Figure 3. Taylor diagram of the climate mean state of each output variable from 2002 to 2004 between the model run with optimal parameters and the CNTL run. 1 in the diagram stands for EXP, and 2 stands for CNTL.**



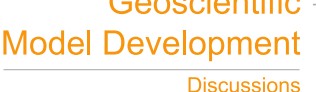

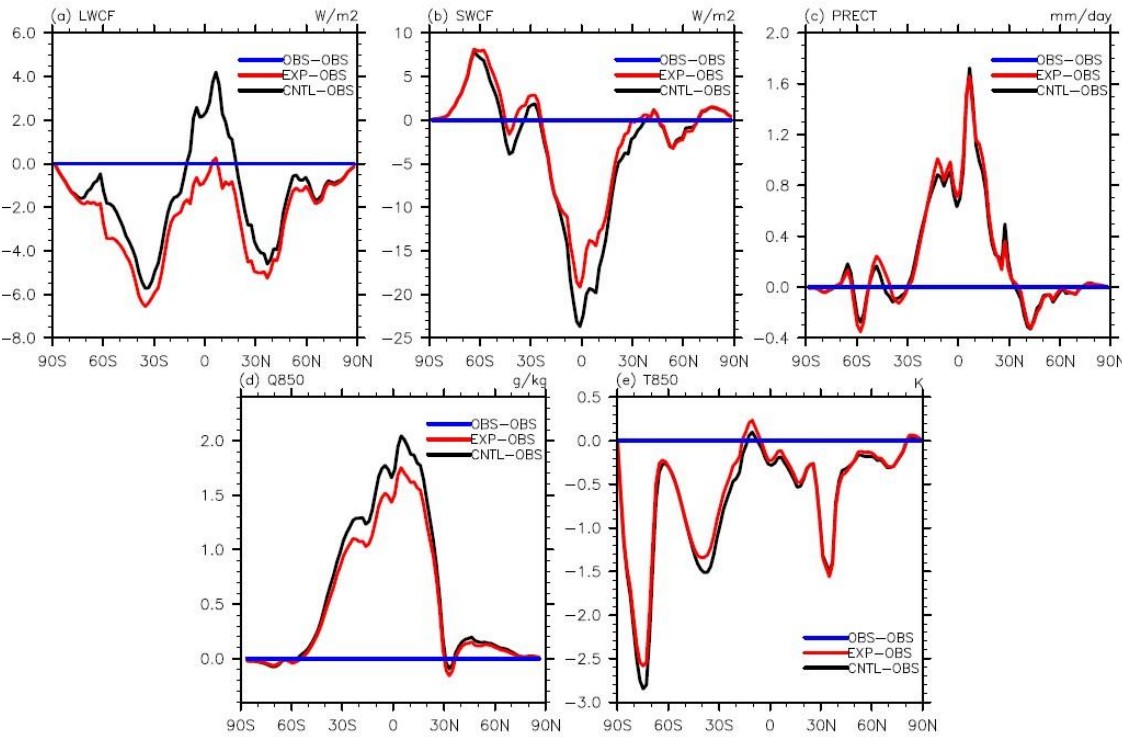

**Figure 4. Meridional distribution of the difference between EXP/CNTL and observed data of (a) LWCF, (b) SWCF, (c) PRECT, (d) Q850, and (e) T850. The position of the dark blue line is 0, the red and black solid lines represent the difference between EXP/ CNTL and the observations.**





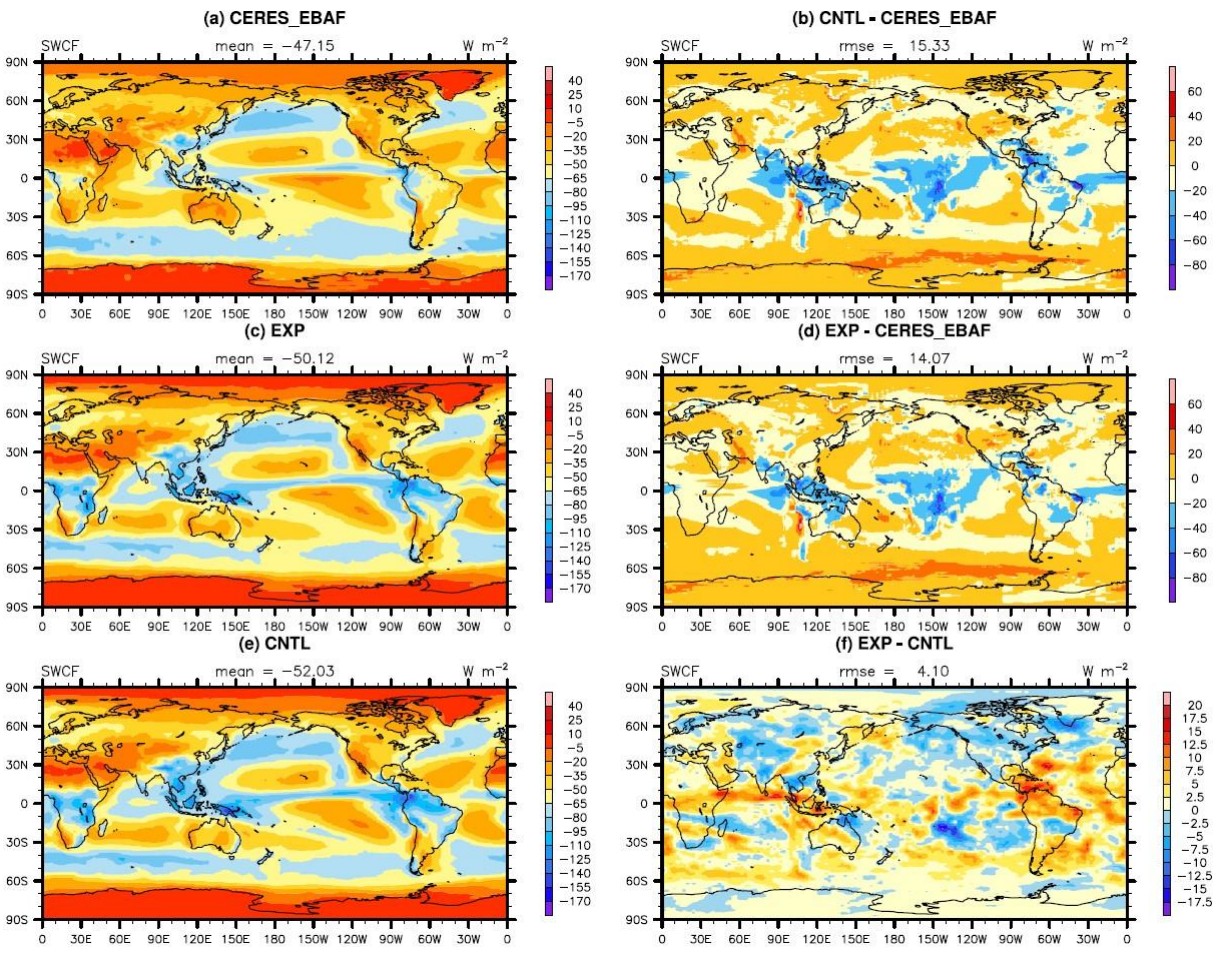

**Figure 5. The spatial distribution of TOA SW cloud forcing of (a) observation, (b) CNTL- observation, (c) EXP, (d) EXP - observation, (e) CNTL and (f) EXP - CNTL.**







**Figure 6. The spatial distribution of specific humidity at 850hPa of (a) observation, (b) CNTL - observation, (c) EXP, (d) EXP - observation, (e) CNTL and (f) EXP - CNTL.**





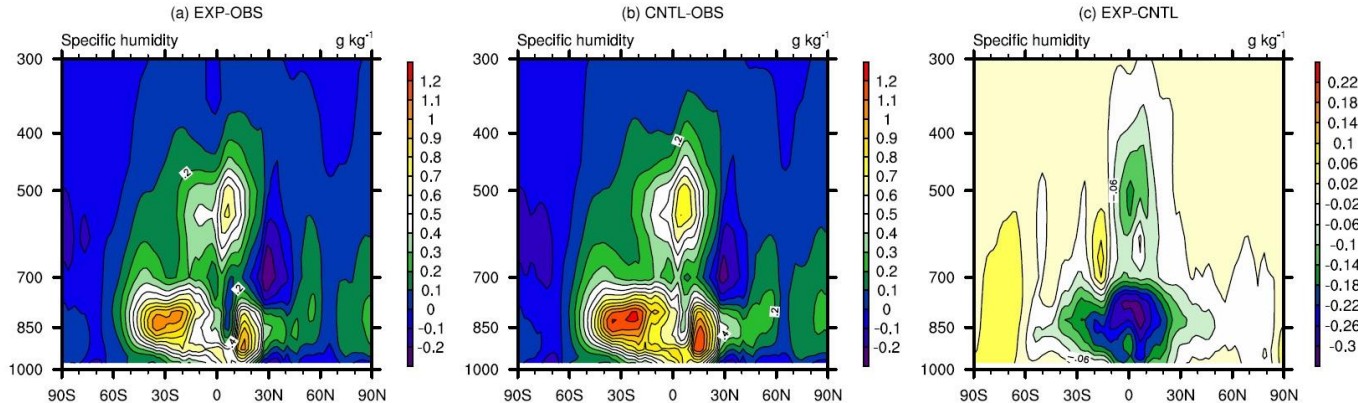

**Figure 7. Pressure-latitude distributions of specific humidity of (a) EXP - OBS, (b) CNTL - OBS and (c) EXP - CNTL.**

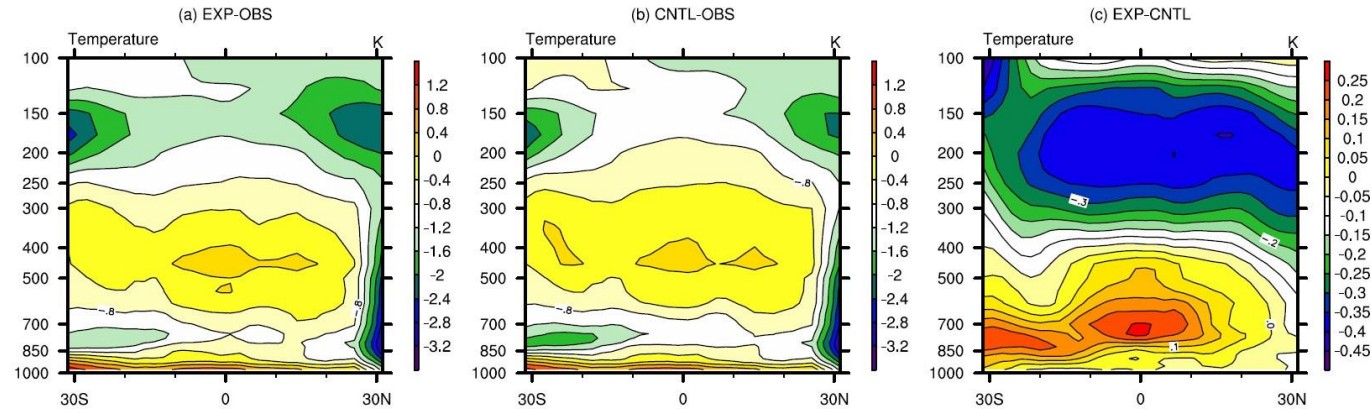

**Figure 8. Pressure-latitude distributions of temperature of (a) EXP - OBS, (b) CNTL - OBS and (c) EXP - CNTL.**



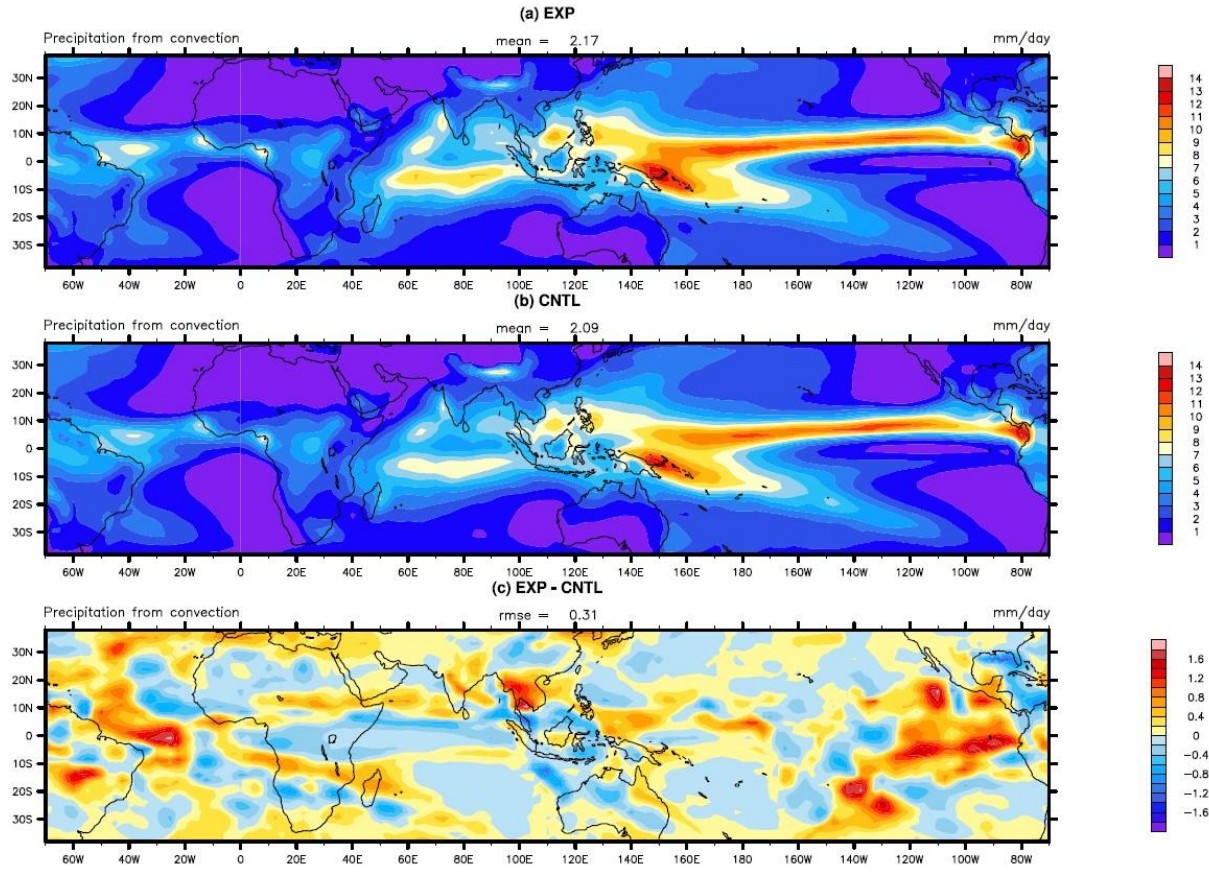

**Figure 9. The spatial distribution of convective precipitation over tropics of EXP (a), CNTL (b), and EXP – CNTL (c).**

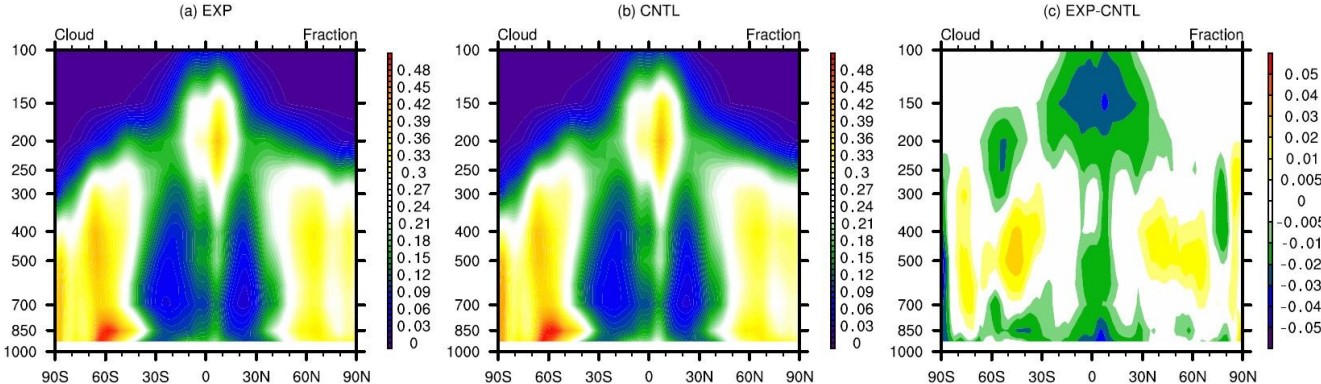





**Figure 10. Pressure-latitude distributions of cloud fraction of EXP (a), CNTL (b), and EXP – CNTL (c).**

**Table 1. Parameters description of CAM5.The default, final optimal values by constrained and unconstrained calibrations, as well as the ranges of parameters. CAPE means the convective available potential energy.**

| Parameter | Description | Range | Default | Unconstrained tune | Constrained tune |
|---|---|---|---|---|---|
| zmconv_c0_lnd | Deep convection precipitation efficiency over land | 2.95e-3 ~ 8.85e-3 | 0.0059 | 0.00319 | 0.00295 |
| zmconv_c0_ocn | Deep convection precipitation efficiency over ocean | 2.25e-2 ~ 6.75e-2 | 0.045 | 0.025 | 0.0225 |
| zmconv_tau | Timescale for consumption rate deep CAPE | 1800 ~ 5400 | 3600 | 1838.814 | 1800 |
| cldfrc_rhminh | Threshold relative humidity for high stable clouds | 0.6 ~ 0.9 | 0.80 | 0.897 | 0.900 |
| cldfrc_rhminl | Threshold relative humidity for low stable clouds | 0.8 ~ 0.95 | 0.8875 | 0.930 | 0.900 |
| cldsed_ai | Fall speed parameter for cloud ice | 300 ~ 1100 | 700 | 853.207 | 970.613 |

5  **Table 2. The output variables used to evaluate performance metric index and the source of the corresponding observations**

| Variable | Full name | OBS |
|---|---|---|
| LWCF | Longwave cloud forcing | CERES-EBAF |
| SWCF | Shortwave cloud forcing | CERES-EBAF |
| PRECT | Total precipitation rate | GPCP |
| Q850 | Specific humidity at 850hPa | ERA-Interim |
| T850 | Temperature at 850hPa | ERA-Interim |

**Table 3 Synthesized performance metric index and radiation bias in the CNTL run, and the optimal model run with unconstrained and constrained methods.**

| | CNTL | Unconstrained tune | Constrained tune |
|---|---|---|---|
| Metric index | 1 | 0.890 | 0.937 |
| Radiation bias | 0.601 | 3.796 | 0.100 |



**Table 4 Performance metric index of each variable in the optimal model run with unconstrained and constrained methods.**

| Variable | Constrained tune |
|----------|------------------|
| LWCF | 1.072 |
| SWCF | 0.841 |
| PRECT | 1.080 |
| Q850 | 0.754 |
| T850 | 0.936 |

5  **Table 5 The percentage of standard deviation of the 8 fields between the CNTL run and the optimal model run with constrained optimization according to the corresponding observations.**

| Standard deviation \|\|%\|\| | Default | Constrained optimization |
|-------------------------------|---------|--------------------------|
| Sea level pressure (ERA-Interim) | 1.124 | 1.053 |
| Land rainfall (30º N–30º S, GPCP) | 0.954 | 0.896 |
| Ocean rainfall (30º N–30º S, GPCP) | 1.283 | 1.236 |
| Land 2m temperature (Willmott) | 1.071 | 1.055 |
| Pacific surface stress (5º N-5º S, ERS) | 1.391 | 1.397 |
| Zonal wind (300 mb, ERA-Interim) | 1.042 | 1.037 |
| Relative humidity (ERA-Interim) | 1.217 | 1.219 |
| Temperature (ERA-Interim) | 1.158 | 1.141 |

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
