# Peer review of "An effective parameter optimization with radiation balance constraint in CAM5 (version 5.3)"

_Geoscientific Model Development, 2019_

## Referee Comment (RC1) · Anonymous Referee #1 · 29 Jul 2019

This study applied an automated parameter optimization algorithm subject to TOA radiation balance constraint to improve the performance of the Community Atmospheric Model in climate simulations. Results showed that the optimized parameters evidently improve the model performance while the energy balance principle can always hold across the entire optimization iterations. This paper conforms the importance of radiation balance constraint for optimization applications in climate models. The manuscript is well organized and the presentation is generally good. However, there are some aspects need to be improved before considering of publication.

Recommendation: Minor revision

Specific comments: 1. The optimization results using the constrained algorithm are quite different from the unconstrained results (Fig. 2). Does this indicate that the bet-

ter model performance based on the synthesized metric (eq. 3) often leads to more serious radiation imbalance at TOA? This issue might be related to the structural inadequacy in the model physics as discussed in Qian et al. (2018) and Yang et al. (2019).

2. The penalty term applied in the cost function (eq. 6) is a key element of the optimization method the authors presented here. I am wondering what the optimization results will be if the net radiation budgets at TOA are directly included in the synthesized metric that is used for optimization. I think by doing this, the best members would be located in some areas between the red and black markers in Fig. 2. The authors can check the results by using the experiments that have already been completed with constrained or unconstrained algorithm.

3. P2L31-32, please check the grammar.

4. P3L18, "into to"?

5. P4L15, "it has been identified as the second most influential parameter in climate", second most influential parameter for which aspects of climate?

6. P4L22, is 1.9*1.9 a standard option of resolution in CAM5? F19 should correspond to a resolution of 1.9*2.5.

7. P4L26, The synthesized metric was based on MSE, while the abstract (i.e. P1L15) said it used global mean values. Please make the statements consistent.

8. Eq. 3, outputs from the control run were used to normalize model errors for different variables. So will the optimization results be different if a different set of parameter values were used in the default configuration?

9. P6L10, "leading" or "leading to"?

10. P7L7, "When the time scale is shorter with unchanged cloud bottom convective mass flux", what is the meaning of "unchanged cloud bottom convective mass flux"?

[Figure]

Shorter time scale should lead to stronger mass flux at cloud base.

Reference Qian, Y., et al. (2018). Parametric Sensitivity and Uncertainty Quantification in the Version 1 of E3SM Atmosphere Model Based on Short Perturbed Parameter Ensemble Simulations, J. Geophys. Res. Atmos., 123, 13046, 2018. Yang, B. et al. (2019).Parametric and structural sensitivities of turbine-height wind speeds in the boundary layer parameterizations in the Weather Research and Forecasting model. J. Geophys. Res. Atmos, 124. https://doi.org/10.1029/2018JD029691
* * *

---

## Referee Comment (RC2) · Anonymous Referee #2 · 7 Aug 2019

This manuscript describes an optimization method to improve the calibration of adjustable parameters in global climate models. This works builds upon previous works by Zhang et al. (2015, 2018). The main difference is the addition of a global constraint to enforce that the net energy imbalance at TOA be less than 1 W/m2. This constraint is incorporated by simply adding a penalty term to the cost function (Eq. 6).

When applied to CAM5.3, the proposed method results in a modest overall improvement of 6.3% in the cost function. Among the fields subject to optimization (LWCF, SWCF, PRECT, Q850, T850), the largest improvements occur for SWCF, Q850, with minor improvement for T850 and minor degradations for LWCF and PRECT (Table 4). Since CAM5.3 is already a well tuned model, it is not particularly surprising the the overall improvement is small.

Overall, the manuscript is clear and easy to read and fits well within the scope of GMD. I would recommend publication after some modifications to further improve it.

* It should be noted that the idea of including a constraint on the global value of net radiation is not new. From Jackson et al. 2008 (J Climate): "We also included a term constraining the global net radiative balance at the top of the atmosphere. We had intended to give this a target of 0.3 W m−2"

* Figure 2 and corresponding text. There is a clear separation between optimized results with and without constraint. This is interesting and warrants further discussion. How different are the unconstrained optimized simulations compared to the constrained ones? This could be illustrated by showing a few selected figures. Also, the constraint is applied as a rather brutal all-or-nothing penalty function that may prevent a wider exploration of the parameter space. One wonders whether a smoother penalty function for the global net radiation have led to different (better) constrained solutions? I would recommend exploring alternate formulations for the penalty function (for example quadratic or exponential) to check whether the specific formulation of the penalty function has any impact on the results.

* Table 1 and corresponding text. Under constrained optimization, the final value for 3 out of 6 parameters hit the lowest allowable limit. This should be discussed.

Minor:

* Page 1, lines 17-18: rephrase to make it clear that the constraint is abs(FLNT-FSNT) < 1.

* Page 1, line 20: "under the premise of a profound understanding": delete. I don't see any new "profound understanding" emerging from this work or method.

* Page 1, line 25: "may result in breaking physical mechanisms that models have to address": delete or clarify what is meant by this (i.e. be specific, not vague).

* Page 2, line 13: "by using" → "using"

* Page 3, line 5: "extreme": delete

* Page 3, lines 10-11: "Qian et al. (2015) indicated that some parameters in cloud microphysics and convection are very sensitive to net radiation flux": isn't this the other way around? Net radiation flux is very sensitive to cloud microphysics and convection parameters.

* Page 7, line 1: "The CNTL experiment has excelled in simulating the spatial distribution of SWCF (Fig. 5c)". With RMSE between 14 and 15 W/m2, neither EXP nor CNTL can realistically be described as excelling in representing SWCF. These are much larger errors than seen in recent CMIP6 models.
* * *

---

## Short Comment (SC1) · 14 Aug 2019

Thank you for putting your analysis code online. Currently you have provided a GitHub link to this code. While GitHub is an excellent development environment, it is not an archival location, and indeed GitHub themselves tell you that you should use Zenodo to archive code for citation in publications[1].

In order to bring the code availability section into conformance with GMD policy[2], please archive the code (using the GitHub Zenodo integration is by far the best way to do this), and cite the resulting reference.
* * *
[1]https://guides.github.com/activities/citable-code/

[2]https://www.geoscientific-model-development.net/about/code_and_data_policy.html

---

## Author Comment (AC1) · 24 Sep 2019

Dear Editor,

Thank you for your reminder about code availability. We created the DOI of the code using zenodo. And in the "Code and data availability" part of our paper, we changed the address of the code to https://doi.org/10.5281/zenodo.3405619.

Sincerely Li Wu, Tao Zhang, Yi Qin and Wei Xue

---

## Author Comment (AC3) · 24 Sep 2019

**Author Response to Reviewer 2**

We thank the reviewer for her/his thoughtful and detailed comments, which helped improve this manuscript. Please note: Reviewer's comments are listed in blue fonts. And responses are italicized text below each reviewer's comments.

This manuscript describes an optimization method to improve the calibration of adjustable parameters in global climate models. This works builds upon previous works by Zhang et al. (2015, 2018). The main difference is the addition of a global constraint to enforce that the net energy imbalance at TOA be less than 1 W/m2. This constraint is incorporated by simply adding a penalty term to the cost function (Eq. 6). When applied to CAM5.3, the proposed method results in a modest overall improvement of 6.3% in the cost function. Among the fields subject to optimization (LWCF, SWCF, PRECT, Q850, T850), the largest improvements occur for SWCF, Q850, with minor improvement for T850 and minor degradations for LWCF and PRECT (Table 4). Since CAM5.3 is already a well-tuned model, it is not particularly surprising the overall improvement is small.

Overall, the manuscript is clear and easy to read and fits well within the scope of GMD.

I would recommend publication after some modifications to further improve it.

*Thank you very much for your recognition. We also appreciate your following comments and suggestions.*

1. It should be noted that the idea of including a constraint on the global value of net radiation is not new. From Jackson et al. 2008 (J Climate): "We also included a term constraining the global net radiative balance at the top of the atmosphere. We had intended to give this a target of 0.3 W m−2".

*Reply: Thank you for the recommended paper. In this paper, the simulation skills of "Net longwave top", "Net shortwave top" and other variables are added to the cost function. Our paper presents the optimization algorithm to strengthen radiation balance as a strong constraint. Meanwhile, other variables are optimized as much as possible if the constraint is met. Of course, both methods are the ideas to the problem, and the papers you recommend also give us some possible directions for future work.*

2. Figure 2 and corresponding text. There is a clear separation between optimized results with and without constraint. This is interesting and warrants further discussion.

How different are the unconstrained optimized simulations compared to the constrained ones? This could be illustrated by showing a few selected figures. Also, the constraint is applied as a rather brutal all-or-nothing penalty function that may prevent a wider exploration of the parameter space. One wonders whether a smoother penalty function for the global net radiation have led to different (better) constrained solutions? I would recommend exploring alternate formulations for the penalty function (for example quadratic or exponential) to check whether the specific formulation of the penalty function has any impact on the results.

*Reply: Nice insight! The reasons why the points of constrained optimization and unconstrained optimization are separated in Figure 2 as follows: The first reason is that we only selected the*

*top 15 optimization results for display. The other points in the tuning process are not so distinct. The second reason is the starting points for optimization we chose leads to the current results in Figure 2. If we use different starting points, the optimization results may be different. The third reason may be also related to your second point. Since we use radiation balance as a particularly strong constraint here, the exploration space in the tuning process tends to be in the space which obeys the constraint. Compared with the unconstrained optimization algorithm, the searching path for optimization of our algorithm is different.*

*You mentioned that the choice of smooth constraints will also have a great impact on the search space, and whether the final optimization results is better or not need to be evaluated carefully. Anyway, it is a very good idea! Sorry, due to the long running time of the AMIP experiment, we can't immediately give the corresponding experimental results. In addition, your proposed quadratic and index constraint forms give us a lot of inspiration. In the future work, we are also about to carry out corresponding experiments.*

3. Table 1 and corresponding text. Under constrained optimization, the final value for 3 out of 6 parameters hit the lowest allowable limit. This should be discussed.

*Reply: Thanks for pointing this out! Indeed, three parameters (zmconv_c0_lnd, zmconv_c0_ocn, zmconv_tau) hit the lowest allowable limit.*

*In CTL experiment, the net TOA imbalance is around 0.6 W/m2, and the incoming shortwave radiation is larger than the outgoing longwave radiation. First, we found that TOA LW radiation and LWCF cannot reach the better performance at the same time, due to the bias in clear-sky longwave radiation flux (FLNTC). So, the radiation balance is a strict constraint, and the performance of LWCF has to be sacrificed. This is revealed by the degraded LWCF performance (1.072) as in Table 4. Second, in the tuning process, we found that shortwave radiation flux is more sensitive to the tuning parameters than longwave radiation flux. To reach a strict small TOA imbalance, the easier tuning direction is to reduce the incoming shortwave radiative flux and to get closer to the outgoing LW radiation flux. Indeed, the final constrained tuning result gets a small TOA imbalance (0.1 W/m2) with TOA shortwave and longwave radiation flux 236.47 W/m2 and 236.37 W/m2, respectively. Three parameters hitting the lowest allowable limit all are used to reduce the incoming shortwave radiation flux largely. To get the final TOA balance and keep an acceptable model performance, the picked tuning parameters here has to hit the lowest limit. It also suggests the difficulty to get perfect performance in all perspectives.*

*We try to discuss this in Section "Interpretation of the results" (Page 7 Line 26-Page 8 Line 2) of our revised version.*

4. Page 1, lines 17-18: rephrase to make it clear that the constraint is abs(FLNT-FSNT) < 1.

*Reply: Thanks for pointing this out. The sentence in lines 17-18 has been revised to "The radiation constraint is defined as the absolute difference between the net longwave flux at top of model (FLNT) and the net solar flux at top of model (FSNT) less than 1 W m-2."*

5. Page 1, line 20: "under the premise of a profound understanding": delete. I don't see any new "profound understanding" emerging from this work or method.

*Reply: The sentence has been deleted.*

6. Page 1, line 25: "may result in breaking physical mechanisms that models have to address": delete or clarify what is meant by this (i.e. be specific, not vague).

*Reply: Thanks. We have deleted the sentence "and may result in breaking physical mechanisms that models have to address."*

7. Page 2, line 13: "by using" → "using"

*Reply: Corrected*

8. Page 3, line 5: "extreme": delete

*Reply: Deleted.*

9. Page 3, lines 10-11: "Qian et al. (2015) indicated that some parameters in cloud microphysics and convection are very sensitive to net radiation flux": isn't this the other way around? Net radiation flux is very sensitive to cloud microphysics and convection parameters.

*Reply: Sorry, this is a mistake. We have modified this sentence to "Net radiation flux is very sensitive to cloud microphysics and convection parameters (Qian et al.2015)."*

10. Page 7, line 1: "The CNTL experiment has excelled in simulating the spatial distribution of SWCF (Fig. 5c)". With RMSE between 14 and 15 W/m2, neither EXP nor CNTL can realistically be described as excelling in representing SWCF. These are much larger errors than seen in recent CMIP6 models.

*Reply: Thanks for pointing this out. The model we used in this work is CAM5.3 with a resolution of 1.9\*2.5, which is different from the atmospheric component of the latest CMIP6 model CESM2. And the resolution is also different. The CESM model of CMIP6 is better than the CESM model used in CMIP5, especially for the simulation results of SWCF. As reported, the CMIP6 model have greatly improved the simulation of SWCF (The global average RMSE is reduced by approximately 5W/m2). However, as we worked on this, the CMIP6 model has not yet been released publicly and we cannot use this improved version to evaluate our tuning algorithm. But we believe that we can also get better performance for the latest model version while keeping the radiation balance. We will try to use the latest model version to the following work.*

---

## Author Response (AR1)

Dear editor,

We thank you and the two reviewers for constructive comments that help us to improve the manuscript. Below we include the detailed, point-by-point response to each comment and question raised by each referee. Please note: Reviewer's comments are listed in blue fonts. And responses are *italicized text* below each reviewer's comments. In our responses, line and page numbers correspond to the enclosed annotated manuscript and annotated supporting information. In the annotated files, red text was added, and  was removed.

In addition, since github is not an archival location, we created the DOI of the code using zenodo. And in the "Code and data availability" part of our revised paper, we changed the address of the code to https://doi.org/10.5281/zenodo.3405619.

Yours sincerely,
Wei Xue

**Author Response to Reviewer 1**

This study applied an automated parameter optimization algorithm subject to TOA radiation balance constraint to improve the performance of the Community Atmospheric Model in climate simulations. Results showed that the optimized parameters evidently improve the model performance while the energy balance principle can always hold across the entire optimization iterations. This paper conforms the importance of radiation balance constraint for optimization applications in climate models. The manuscript is well organized and the presentation is generally good.

*Thank you very much for your recognition. We also appreciate your following comments and suggestions.*

However, there are some aspects need to be improved before considering of publication.
1. The optimization results using the constrained algorithm are quite different from the unconstrained results (Fig. 2). Does this indicate that the better model performance based on the synthesized metric (eq. 3) often leads to more serious radiation imbalance at TOA? This issue might be related to the structural inadequacy in the model physics as discussed in Qian et al. (2018) and Yang et al. (2019).

*Reply: No. The better model performance based on this synthesized metric does not necessarily lead to serious radiation imbalance problems, which depends on the specified model as well as the parameter impacts and sensitivities. For example, the same metrics used in Zhang et al. (2015). The best parameter configuration in the paper can not only improve the model performance of the Grid-point Atmospheric Model of IAP LASG version 2 (GAMIL2), but also ensure the radiation balance. However, our experimental data on CAM5 shows that the best parameter configuration of the model is very likely to introduce radiation imbalance. This may indicate that it is difficult to optimize multiple variables under radiation balance constraint in a well-tuned model, due to the structural inadequacy in the model physics, as mentioned in Qian et al. (2018). And Yang et al. (2019) also indicated that the structural and parametric*

*problems associated with physical parameterizations are often tied together in weather and climate models. The difficulty of simultaneous optimization of multiple variables also highlights the need of characterizing model structural uncertainty. Moreover, we add the recent paper Yang et al. (2019) in our reference list (p7, line31).*

2. The penalty term applied in the cost function (eq. 6) is a key element of the optimization method the authors presented here. I am wondering what the optimization results will be if the net radiation budgets at TOA are directly included in the synthesized metric that is used for optimization. I think by doing this, the best members would be located in some areas between the red and black markers in Fig. 2. The authors can check the results by using the experiments that have already been completed with constrained or unconstrained algorithm.

*Reply: Thanks for your comments. It is critical for optimization results. In this work, our idea is to use radiation balance as a strong constraint, since only when it is satisfied the model can run stably for a long time. If the net radiation budgets at TOA are directly put into metrics, it is possible that the overall performance indicators will be improved, but the radiation is still unbalanced. The importance of radiation balance has not been emphasized. As for the tuning process, the searching path for optimization with the method the reviewer suggested is different from that of our algorithm (as mentioned in the review), and the final optimization results need to be evaluated carefully. In the future work, we will continue to pay attention to the impact of different constraint forms. For example, the multi-object optimization method by using radiation constraints as one separated optimization object, or using a smoother constraint expression (such as an index or a quadratic expression) and designing smart searching strategies to avoid the balance-broken optimized results.*

3. P2L31-32, please check the grammar.
*Reply: We change it to "This paper takes radiation balance as an example. According to the Earth's energy conservation theory, the absorbed solar radiation is approximately equal to outgoing longwave radiation at the top of model."(p2, line31).*

4. P3L18, "into to"?
*Reply: We change it to "into" (p3, line18).*

5. P4L15, "it has been identified as the second most influential parameter in climate", second most influential parameter for which aspects of climate?
*Reply: The cloud ice sedimentation velocity has been identified as the second most influential parameter in climate sensitivity experiment, in which the simulated performance of Surface Temperature (TAS), Seasonal Cycle in TAS (JJA - DJF), SW upward radiation at TOA, LW upward radiation at TOA, Total Precipitation, etc. is used as the criterion (Sanderson et al., 2008). This sentence is revised as follows: "it has been identified as a high-impact parameter in sensitivity experiments related to temperature, radiation, and precipitation, etc. (Sanderson et al., 2008)." (p4, line15-16).*

6. P4L22, is 1.9*1.9 a standard option of resolution in CAM5? F19 should correspond to a resolution of 1.9*2.5.

*Reply: 1.9\*2.5 is a standard option of resolution in CAM5. We are very sorry for this mistake. Thanks for pointing out this problem. And we have corrected the description of the resolution to 1.9\*2.5. (p4, line23).*

7. P4L26, The synthesized metric was based on MSE, while the abstract (i.e. P1L15) said it used global mean values. Please make the statements consistent.

*Reply: Thanks for pointing out this problem. We change abstract to "in terms of a synthesized performance metric using normalized mean square error of radiation" (p1, line15)*

8. Eq. 3, outputs from the control run were used to normalize model errors for different variables. So will the optimization results be different if a different set of parameter values were used in the default configuration?

*Reply: The model performance optimization percentage of this paper is relative to the current default experiment. If the default parameter configuration is changed, the percentage of performance improvement may be different. The intent of the optimization method in this paper is to further improve the performance of the model under the current configuration conditions. This uncertain parameter optimization method generally improves the performance of the model by about 5% to 10% under the default parameters provided when the model was released. For example, we have increased the radiation balance constraint here, and the performance of CAM5 has been improved by about 6%. In Zhang et al. (2015)'s paper, GAMIL2 performance have been enhanced by about 9%. These results prove that the effectiveness of our optimization method is not accidental. Especially for a new model, our method can help the model experts to find a set of better parameters and can also promote the development and improvement of the model.*

9. P6L10, "leading" or "leading to"?
*Reply: We change it to "leading to". (p6, line 10).*

10. P7L7, "When the time scale is shorter with unchanged cloud bottom convective mass flux", what is the meaning of "unchanged cloud bottom convective mass flux"? Shorter time scale should lead to stronger mass flux at cloud base.

*Reply: Nice insight! Yes, the shorter time scale should lead to stronger cloud-base mass flux based on the closure in deep convection parameterization. We are sorry for this mistake, and what we were going to state is that shorter time scale with unchanged CAPE would lead to the stronger cloud-base mass flux. This sentence has been revised as follows: "When the time scale is shorter with less changed CAPE, the increased cloud-base mass flux would help to enhance the convective precipitation." (p7, line7-9).*

How different are the unconstrained optimized simulations compared to the constrained ones? This could be illustrated by showing a few selected figures. Also, the constraint is applied as a rather brutal all-or-nothing penalty function that may prevent a wider exploration of the parameter space. One wonders whether a smoother penalty function for the global net radiation have led to different (better) constrained solutions? I would recommend exploring alternate formulations for the penalty function (for example quadratic or exponential) to check whether the specific formulation of the penalty function has any impact on the results.

*Reply: Nice insight! The reasons why the points of constrained optimization and unconstrained optimization are separated in Figure 2 as follows: The first reason is that we only selected the top 15 optimization results for display. The other points in the tuning process are not so distinct. The second reason is the starting points for optimization we chose leads to the current results in Figure 2. If we use different starting points, the optimization results may be different. The third reason may be also related to your second point. Since we use radiation balance as a particularly strong constraint here, the exploration space in the tuning process tends to be in the space which obeys the constraint. Compared with the unconstrained optimization algorithm, the searching path for optimization of our algorithm is different.*

*You mentioned that the choice of smooth constraints will also have a great impact on the search space, and whether the final optimization results is better or not need to be evaluated carefully. Anyway, it is a very good idea! Sorry, due to the long running time of the AMIP experiment, we can't immediately give the corresponding experimental results. In addition, your proposed quadratic and index constraint forms give us a lot of inspiration. In the future work, we are also about to carry out corresponding experiments.*

3. Table 1 and corresponding text. Under constrained optimization, the final value for 3 out of 6 parameters hit the lowest allowable limit. This should be discussed.

*Reply: Thanks for pointing this out! Indeed, three parameters (zmconv_c0_lnd, zmconv_c0_ocn, zmconv_tau) hit the lowest allowable limit.*

*In CTL experiment, the net TOA imbalance is around 0.6 W/m2, and the incoming shortwave radiation is larger than the outgoing longwave radiation. First, we found that TOA LW radiation and LWCF cannot reach the better performance at the same time, due to the bias in clear-sky longwave radiation flux (FLNTC). So, the radiation balance is a strict constraint, and the performance of LWCF has to be sacrificed. This is revealed by the degraded LWCF performance (1.072) as in Table 4. Second, in the tuning process, we found that shortwave radiation flux is more sensitive to the tuning parameters than longwave radiation flux. To reach a strict small TOA imbalance, the easier tuning direction is to reduce the incoming shortwave radiative flux and to get closer to the outgoing LW radiation flux. Indeed, the final constrained tuning result gets a small TOA imbalance (0.1 W/m2) with TOA shortwave and longwave radiation flux 236.47 W/m2 and 236.37 W/m2, respectively. Three parameters hitting the lowest allowable limit all are used to reduce the incoming shortwave radiation flux largely. To get the final TOA balance and keep an acceptable model performance, the picked tuning parameters here has to hit the lowest limit. It also suggests the difficulty to get perfect performance in all perspectives.*

*We added some discussion in Section "Interpretation of the results" of our revised version.*

*"Note that three of six parameters hit their lowest allowable limit with the TOA balance constraint. We found that the incoming shortwave radiation flux is more sensitive to tuning parameters than the outgoing longwave radiation flux. Thus, to reduce the TOA imbalance (SW-LW) and keep the reasonable model performance, the shortwave radiation flux should be reduced largely via increasing low cloud fraction and liquid water content. These three variables hit the lowest bound are the dominant factors. This suggests that getting both the TOA balance and reasonable model performance is a relatively complex and difficult problem, as pointed out by Qian et al. (2018). Meanwhile, how to get picked parameters with similar*

*sensitivity to both longwave and shortwave radiation flux might be a potential approach to overcome the bound limit and it warrants further studies." (p7, line 26-p8, line2).*

4. Page 1, lines 17-18: rephrase to make it clear that the constraint is abs(FLNT-FSNT) < 1.
*Reply: Thanks for pointing this out. The sentence in lines 17-18 has been revised to "The radiation constraint is defined as the absolute difference between the net longwave flux at top of model (FLNT) and the net solar flux at top of model (FSNT) less than 1 W m-2." (p1, line 17-18).*

5. Page 1, line 20: "under the premise of a profound understanding": delete. I don't see any new "profound understanding" emerging from this work or method.
*Reply: The sentence has been deleted (p1, line 20).*

6. Page 1, line 25: "may result in breaking physical mechanisms that models have to address": delete or clarify what is meant by this (i.e. be specific, not vague).
*Reply: Thanks. We have deleted the sentence "and may result in breaking physical mechanisms that models have to address." (p1, line 25).*

7. Page 2, line 13: "by using" → "using"
*Reply: Corrected (p2, line 13).*

8. Page 3, line 5: "extreme": delete
*Reply: Deleted (p3, line 5).*

9. Page 3, lines 10-11: "Qian et al. (2015) indicated that some parameters in cloud microphysics and convection are very sensitive to net radiation flux": isn't this the other way around? Net radiation flux is very sensitive to cloud microphysics and convection parameters.
*Reply: Sorry, this is a mistake. We have modified this sentence to "Net radiation flux is very sensitive to cloud microphysics and convection parameters (Qian et al.2015)." (p3, line 10).*

10. Page 7, line 1: "The CNTL experiment has excelled in simulating the spatial distribution of SWCF (Fig. 5c)". With RMSE between 14 and 15 W/m2, neither EXP nor CNTL can realistically be described as excelling in representing SWCF. These are much larger errors than seen in recent CMIP6 models.
*Reply: Thanks for pointing this out. The model we used in this work is CAM5.3 with a resolution of 1.9\*2.5, which is different from the atmospheric component of the latest CMIP6 model CESM2. And the resolution is also different. The CESM model of CMIP6 is better than the CESM model used in CMIP5, especially for the simulation results of SWCF. As reported, the CMIP6 model have greatly improved the simulation of SWCF (The global average RMSE is reduced by approximately 5W/m2). However, as we worked on this, the CMIP6 model has not yet been released publicly and we cannot use this improved version to evaluate our tuning algorithm. But we believe that we can also get better performance for the latest model version while keeping the radiation balance. We will try to use the latest model version to the following work.*

[revised manuscript text omitted]